# Adsorption Characteristics between Ti Atoms of TiO_2_(100) and Corrosive Species of CO_2_-H_2_S-Cl^−^ System in Oil and Gas Fields

**DOI:** 10.3390/ma16083129

**Published:** 2023-04-16

**Authors:** Shidong Zhu, Ke Wang, Haixia Ma, Pan Dong

**Affiliations:** 1School of Materials Science and Engineering, Xi’an Shiyou University, Xi’an 710065, China; zhusdxt@126.com; 2Shaanxi Key Laboratory of Carbon Dioxide Sequestration and Enhanced Oil Recovery, Shaanxi Yanchang Petroleum (Group) Co., Ltd., Xi’an 710065, China; windsgone@foxmail.com; 3School of Chemical Engineering, Northwest University Shaanxi, Xi’an 710069, China; mahx@nwu.edu.cn

**Keywords:** TiO_2_(100), first principles, thermodynamic stability, adsorption characteristics, corrosive species

## Abstract

The service environment of OCTG (Oil Country Tubular Goods) in oil and gas fields is becoming more and more severe due to the strong affinity between ions or atoms of corrosive species coming from solutions and metal ions or atoms on metals. While it is difficult for traditional technologies to accurately analyze the corrosion characteristics of OCTG in CO_2_-H_2_S-Cl^−^ systems, it is necessary to study the corrosion-resistant behavior of TC4 (Ti-6Al-4V) alloys based on an atomic or molecular scale. In this paper, the thermodynamic characteristics of the TiO_2_(100) surface of TC4 alloys in the CO_2_-H_2_S-Cl^−^ system were simulated and analyzed by first principles, and the corrosion electrochemical technologies were used to verify the simulation results. The results indicated that all of the best adsorption positions of corrosive ions (Cl^−^, HS^−^, S^2−^, HCO_3_^−^, and CO_3_^2−^) on TiO_2_(100) surfaces were bridge sites. A forceful charge interaction existed between Cl, S, and O atoms in Cl^−^, HS^−^, S^2−^, HCO_3_^−^, CO_3_^2−^, and Ti atoms in TiO_2_(100) surfaces after adsorption in a stable state. The charge was transferred from near Ti atoms in TiO_2_ to near Cl, S, and O atoms in Cl^−^, HS^−^, S^2−^, HCO_3_^−^, and CO_3_^2−^. Electronic orbital hybridization occurred between 3p^5^ of Cl, 3p^4^ of S, 2p^4^ of O, and 3d^2^ of Ti, which was chemical adsorption. The effect strength of five corrosive ions on the stability of TiO_2_ passivation film was S^2−^ > CO_3_^2−^ > Cl^−^ > HS^−^ > HCO_3_^−^. In addition, the corrosion current density of TC4 alloy in different solutions containing saturated CO_2_ was as follows: NaCl + Na_2_S + Na_2_CO_3_ > NaCl + Na_2_S > NaCl + Na_2_CO_3_ > NaCl. At the same time, the trends of *R*_s_ (solution transfer resistance), *R*_ct_ (charge transfer resistance), and *R*_c_ (ion adsorption double layer resistance) were opposite to the corrosion current density. The corrosion resistance of TiO_2_ passivation film to corrosive species was weakened owing to the synergistic effect of corrosive species. Severe corrosion resulted, especially pitting corrosion, which further proved the simulation results mentioned above. Thus, this outcome provides the theoretical support to reveal the corrosion resistance mechanism of OCTG and to develop novel corrosion inhibitors in CO_2_-H_2_S-Cl^−^ environments.

## 1. Introduction

Corrosion has been considered as one of the major social problems in pipelines and industries using such materials since the early industrial revolution. A large number of accidents occur frequently, which leads to a greater threat to the safe production of oil and gas [1]. The annual cost of corrosion in China is about CNY 2.3 trillion, accounting for 3.3% of GDP [2].

In recent years, with the development of deep and ultra-deep wells to meet the social demand for energy, the working environment of tubing and casing is becoming more and more complex. In addition to the stringent service conditions, metal OCTG are inevitably subjected to different degrees of corrosion, and the working properties of OCTG have decreased. For example, the presence of H_2_S leads to severe localized corrosion, as well as cracks caused by stress and hydrogen [3]. In some special working conditions, CO_2_ and H_2_S exist at the same time [4], which greatly deteriorates the service environment of OCTG [5]. The high temperature, pressure, acid gas content, and Cl^−^ concentration of oil and gas wells increase the requirements for corrosion-resistant OCTG.

The TC4 titanium alloy (Ti-6Al-4V) is now considered to be the ideal material applied in oil and gas fields, accounting for about half of the market share of titanium alloys currently used in the world [6]. A dense TiO_2_ oxide film with a thickness of 4~6 nm of TC4 will be spontaneously formed at room temperature [7], which can effectively prevent the matrix in a solution from being corroded by corrosive ions (such as H^+^, Cl^−^, etc.) [8]. However, the film is not always able to maintain its integrity; it is very likely to be destroyed in some medium containing some corrosive species, resulting in serious corrosion of the titanium alloy matrix [9]. It is difficult for traditional technologies to accurately analyze the corrosion characteristics of OCTG in CO_2_-H_2_S-Cl^−^ systems.

Therefore, the first-principles calculation software (Materials Studio) on account of DFT (Density Functional Theory) was selected to research the interface characteristics between the corrosive species and TiO_2_ passivation film on the surface of TC4 alloys in CO_2_-H_2_S-Cl^−^ systems containing Cl^−^, HS^−^, S^2−^, HCO_3_^−^, and CO_3_^2−^ based on an atomic or molecular scale. Additionally, the corrosion characteristics of the TC4 alloy in NaCl, NaCl + Na_2_CO_3_, NaCl + Na_2_S, and NaCl + Na_2_S + Na_2_CO_3_ solutions containing saturated CO_2_ were carried out by the electrochemical technologies to verify the simulation results above.

## 2. Research Methods

### 2.1. First Principles

#### 2.1.1. Modeling

TiO_2_ passivation film on titanium alloy surfaces has three crystal structures: rutile, anatase, and brookite [10]. Figure 1 shows the Raman spectra of TiO_2_ film on TC4 alloy, and Table 1 shows the frequency shift positions of Raman spectral characteristics of three crystalline TiO_2_. The four peaks, 145.53 cm^−1^, 241.76 cm^−1^, 612.53 cm^−1^, and 824.16 cm^−1^ in Figure 1, are consistent with the corresponding peak value of the rutile TiO_2_ in Table 1. Some scholars found that the composition of titanium alloy passivation film was rutile TiO_2_ [11]. Therefore, rutile TiO_2_ was selected as the research object in this paper.

There are (110), (100), and (001) low index surfaces in Rutile phase TiO_2_. The characteristics of the various ions on TiO_2_(110) surfaces have been studied, including our previous research [2], but few reports were focused on the adsorption of TiO_2_(100) and TiO_2_(001) surfaces. Furthermore, compared with TiO_2_(001) surfaces, TiO_2_(100) surfaces present a higher possibility of stable existence at high temperatures [12]. Therefore, the adsorption properties of various corrosive ions (Cl^−^, HS^−^, S^2−^, HCO_3_^−^, and CO_3_^2−^) on rutile TiO_2_(100) surfaces were studied.

The CASTEP in Material Studio, the first-principles computing software, was used to conduct geometric optimization for all adsorption configurations [13]. According to the setting requirements of the CASTEP module, a 2 × 3 × 1 three-dimensional supercell structure with periodic boundary conditions was established for the rutile TiO_2_(100) surface. In addition, a vacuum area with a thickness of 20 Å was added between the two plates to prevent interaction between them [14]. Figure 2 reveals the boundary surface models of various corrosive ions (Cl^−^, HS^−^, S^2−^, HCO_3_^−^, and CO_3_^2−^) at different adsorption sites (top, bridge, and hole) on a TiO_2_(100) surface.

#### 2.1.2. Computing Method

Using the PBE functional of GGA, the pseudopotentials were constructed using the plane wave ultrasoft pseudopotential SCF [1,12,15], where the truncation energy of the plane wave was set as 400 eV, the convergence accuracy in the iteration process was 2 × 10^−6^ eV/atom, the self-consistent iteration was 300 times, the force converge was 0.03 eV/atom, the tolerance deviation was not higher than 0.005, the stress deviation was under 0.08 GPa, and the k-points value was 2 × 3 × 1 in the Brillouin zone.

### 2.2. Electrochemical Test

#### 2.2.1. Preparation of Experimental Materials

The electrochemical test sample was TC4 titanium alloy, which was ø10 mm × 3 mm. A wire was welded to one end of the sample and tested for conductivity with a multimeter to verify whether the wire was welded correctly. The surface at the other end of the sample was the electrochemical test surface. The surface other than the electrochemical test surface was glued and stamped with epoxy resin AB glue and then polished with sandpaper with mesh sizes of 400^#^, 800^#^, 1200^#^, 1500^#^, and 2000^#^. For the purpose of reaching the test requirements for sample roughness, the sample surface was polished to 2000^#^, cleaned with distilled water, degreased with acetone, dehydrated and desiccated with alcohol, and dried with cold air for later use.

#### 2.2.2. Experimental Methods and Equipment

The electrochemical test was carried out by Princeton P4000 electrochemical workstation, in which the working electrode was TC4 alloy, the reference electrode was polytetrafluoro silver chloride, and the auxiliary electrode was a platinum electrode. Before the electrochemical test, high-purity nitrogen was used to deoxygenate the required corrosive medium for 1 h., and the temperature was heated up to the preset temperature (80 °C). The electrochemical test was performed when the entire test system reached stability, and each experiment was performed three times.

The working electrode was pre-polarized at a voltage set value of −1200 mV for 3 min in advance of the electrochemical test. After the oxide film spontaneously took shape on the surface of the sample in the air and was eliminated, the working electrode was put in the prepared medium and stood for 30 min to form new film. The test frequency was set to 10^−2^ HZ~10^5^ HZ, the measured signal amplitude was 10 mV sine wave, and the number of points was 50. The scanning rat was set as 0.3333 mV/s, and the potential was −1000 mV~+1600 mV.

The corrosive medium was 35 g/L NaCl, 35 g/L NaCl + 1 g/L Na_2_CO_3_, 35 g/L NaCl + 1 g/L Na_2_S, and 35 g/L NaCl + 1 g/L Na_2_CO_3_ + 1 g/L Na_2_S, respectively, which were all chemically pure agents.

## 3. Results and Discussion

### 3.1. Thermodynamic Stability of Passivation Film

#### 3.1.1. Stable Adsorption Model

To simulate the species in the CO_2_-H_2_S-Cl^−^ environment (CO_2_+H_2_O→H_2_CO_3_, H_2_CO_3_→H^+^+HCO_3_^−^, HCO_3_^−^→H^+^+CO_3_^2−^, H_2_S→H^+^+HS^−^, HS^−^→H^+^+S^2−^), the final energy of five corrosive ions (Cl^−^, HS^−^, S^2−^, HCO_3_^−^, and CO_3_^2−^) at different adsorption sites on TiO_2_(100) surface after geometric optimization is shown in Table 2. By comparison, it was found that the energy of each corrosive ion was the lowest at the bridge site of the TiO_2_(100) surface. If the energy of the adsorption system were more negative, its structure would be more stable [16]. Therefore, it can be determined that all of the best adsorption sites of Cl^−^, HS^−^, S^2−^, HCO_3_^−^, and CO_3_^2−^ on the TiO_2_(100) surface were bridge sites. The final energy of each corrosive ion at the bridge site of the TiO_2_(100) surface was in the following order: S^2−^ > HS^−^ > Cl^−^ > CO_3_^2−^ > HCO_3_^−^.

#### 3.1.2. Charge Density

Figure 3 reveals the charge density distribution of each corrosive ion (Cl^−^, HS^−^, S^2−^, HCO_3_^−^, and CO_3_^2−^) at the bridge site of the TiO_2_(100) surface. It could be seen that a forceful charge interaction exists between the Cl, S, O, and Ti atoms which was in the Cl^−^, HS^−^ and S^2−^, HCO_3_^−^, CO_3_^2−^, and TiO_2_(100) surface, respectively.

Table 3 reveals the charge numbers of Cl, S, and O atoms in various corrosive ions. It could be seen that the absolute values are as follows: S^2−^(S) > CO_3_^2−^(O) > Cl^−^(Cl) > HS^−^(S) > HCO_3_^−^(O), which are in accordance with our previous research [2]. The metal surface with a higher charge density value is more likely to be corroded by the corrosive ions, leading to the TiO_2_ passivation film suffering from stronger corrosion. It could be seen that the stability of the TiO_2_ in the environment containing corrosive species was the following: S^2−^ < CO_3_^2−^ < Cl^−^ < HS^−^ < HCO_3_^−^. That is, TiO_2_ film on the surface of TC4 alloy is more easily damaged in the mediums containing S^2−^ than in CO_3_^2−^, Cl^−^, HS^−^, and HCO_3_^−^.

#### 3.1.3. Charge Density Difference

Figure 4 shows the charge density of the corrosive ions (Cl^−^, HS^−^, S^2−^, HCO_3_^−^, and CO_3_^2−^) at the bridge site of the TiO_2_(100) under the stable state adsorption. It could be seen that a very distinct charge transfer appearance was presented between Cl, S, O, and Ti atoms which was in the Cl^−^, HS^−^ and S^2−^, HCO_3_^−^ and CO_3_^2−^, and TiO_2_(100) surfaces, respectively. Charge segregation and electronegativity decreased near Cl, S, and O atoms, while charge dissipation and electronegativity increased near the Ti atom in TiO_2_ [17]. Therefore, the interface binding energy between Cl, S, O, and Ti atoms was in the Cl^−^, HS^−^ and S^2−^, HCO_3_^−^ and CO_3_^2−^, and TiO_2_(100) surfaces, respectively. Finally, the specific charge transfer process moved from the Ti atom on the TiO_2_(100) surface to Cl, S, and O atoms.

#### 3.1.4. Density of States

Figure 5 shows PDOS (Projected Density of States) diagrams of five corrosive ions at the bridge site of the TiO_2_(100) surface, which can be calculated to investigate the characteristics of various ions on the TiO_2_(100) surface deeply [18]. It could be seen that a certain extent of the charge interaction existed between Cl, S, O, and Ti atoms, indicating that the adsorption process was chemical adsorption [19]. The charge interaction and interfacial bonding were primarily made of hybrid orbitals between 3d^2^ of the Ti atoms and 3p^5^ of Cl, 3p^4^ of S, and 2p^4^ of O.

#### 3.1.5. Binding Energy

The corrosiveness of each corrosive ion to the matrix can be ensured through the interface binding energy, which was calculated as follows [20]:(1)$$E=E_{t}-(E_{1}+E_{2})$$

*E*_t_ is the total energy of whole model after geometry optimization; *E*_1_ is the energy after geometric optimization of TiO_2_(100); *E*_2_ is the energy after geometric optimization of each corrosive ion.

Table 4 displays the final energy between Cl^−^, HS^−^, S^2−^, HCO_3_^−^, CO_3_^2−^, and TiO_2_(100) after geometric optimization. According to Table 2 and Table 4, combined with Formula (1), the interface binding energies of various corrosive ions (Cl^−^, HS^−^, S^2−^, HCO_3_^−^, and CO_3_^2−^) at the bridge site of the TiO_2_(100) surface were obtained, as shown in Table 5. It could be seen that when HCO_3_^−^ was adsorbed on the TiO_2_(100) surface, the entire adsorption system had low energy. Compared with Cl^−^, HS^−^, HCO_3_^−^, and CO_3_^2−^, the interface between S^2−^ and TiO_2_(100) was easier to bond and react, indicating that S^2−^ had a stronger adsorption capacity on TiO_2_. Therefore, TiO_2_ has poor stability in the environment containing S^2−^. The steadier the interface model is, the smaller interface binding energy is [20], so the film stability of TiO_2_ in the solutions containing Cl^−^, HS^−^, S^2−^, HCO_3_^−^, and CO_3_^2−^ was S^2−^ < CO_3_^2−^ < Cl^−^ < HS^−^ < HCO_3_^−^, which is consistent with the charge density results mentioned above.

### 3.2. Corrosion Behavior

#### 3.2.1. Alternating-Current Impedance

The alternating-current impedances of TC4 alloy in NaCl, NaCl + Na_2_CO_3_, NaCl + Na_2_S and NaCl + Na_2_S + Na_2_CO_3_ solutions containing saturated CO_2_ are shown in Figure 6. It could be seen that the radius of the capacitive arc of TC4 alloy in four corrosive solution was NaCl > NaCl + Na_2_CO_3_ > NaCl + Na_2_S > NaCl + Na_2_S + Na_2_CO_3_. The radius of electrochemical Nyquist impedance spectroscopy can determine the corrosion resistance of materials; the larger the radius of the electrochemical Nyquist impedance spectrum is, the stronger the corrosion resistance of materials to local corrosion is [21]. Therefore, the corrosiveness of four corrosive solutions to TC4 alloy was NaCl + Na_2_S + Na_2_CO_3_ > NaCl + Na_2_S > NaCl + Na_2_CO_3_ > NaCl.

The equivalent circuit was shown in Figure 7. It can be seen that *C*_dl_ (double layer capacitance) and *C*_c_ (ion adsorption double layer capacitance on the electrode surface) increased, and that both *R*_ct_ (charge transfer resistance) and *R*_c_ (ion adsorption double layer resistance) decreased, concluding that the TC4 alloy has poor corrosion resistance [22].

As seen in Table 6, when there was only NaCl in the electrolyte, the *C*_c_ value of ion adsorption double layer capacitance on the electrode surface was 3.617 × 10^−7^, the *C*_dl_ value of double layer capacitance was 4.925 × 10^−6^, the *R*_c_ value was 1565 Ω·cm^2^, and the *R*_ct_ value was 3.135 × 10^4^ Ω·cm^2^. With the addition of CO_3_^2−^ and S^2−^, the values of *C*_c_ and *C*_dl_ increased to varying degrees, while the values of *R*_c_ and *R*_ct_ decreased. When CO_3_^2−^ and S^2−^ exited together, the corresponding electrochemical parameters increased. The corrosion resistance of the TC4 alloy to four solutions is NaCl > NaCl + Na_2_CO_3_ > NaCl + Na_2_S > NaCl + Na_2_S + Na_2_CO_3_, which is consistent with the above numerical simulation results.

#### 3.2.2. Polarization Curve

Figure 8 displays the polarization curves of the TC4 titanium alloy in four corrosive media (NaCl, NaCl + Na_2_CO_3_, NaCl + Na_2_S, NaCl + Na_2_S + Na_2_CO_3_). Table 7 shows the fitting results. The *i*_corr_ (self-corrosion current density) was 1.689 × 10^−4^ mA/cm^2^, and the *E*_corr_ (self-corrosion potential) of TC4 alloy in NaCl solution containing saturated CO_2_ was −578 mV. With the addition of CO_3_^2−^ or/and S^2−^, the *E*_corr_ of the electrode decreased, and *i*_corr_ increased. The *E*_corr_ can reflect the tendency of corrosion [23], and the *i*_corr_ represents the speed of corrosion rate. The value of the *i*_corr_ is larger, indicating that the corrosion rate is more rapid [24]. It could be seen that the TC4 titanium alloy showed excellent corrosion resistance in a corrosive solution containing only NaCl. In a NaCl + Na_2_CO_3_ solution, the resistance of the TC4 alloy decreased. While in the NaCl + Na_2_S + Na_2_CO_3_ solution, the TC4 alloy suffered from the most severe corrosion. This finding is consistent with the above alternating-current impedance results and numerical simulation results.

The results of the electrochemical experiments mentioned above also are in good accordance with the previous research in a 35% NaCl + 0.4% Na_2_S solution at 80 °C [25], as shown in Table 8.

## 4. Conclusions

(1)All of the most suitable adsorption sites of corrosive ions (Cl^−^, HS^−^, S^2−^, HCO_3_^−^, and CO_3_^2−^) on the TiO_2_(100) surface were bridge sites, then hole sites and top sites.(2)A forceful charge interaction occurred between Cl, S, O, and Ti atoms. The charge was transferred from near the Ti atoms in the TiO_2_(100) surface to near Cl, Ss, and O atoms in Cl^−^, HS^−^, S^2−^, HCO_3_^−^, and CO_3_^2−^, respectively. Interface binding energy was primarily formed by electronic orbital hybridization between 3p^5^ of Cl, 3p^4^ of S, 2p^4^ of O, and 3d^2^ of Ti, and they were chemical adsorption.(3)Interface binding energy between five corrosive species and the TiO_2_(100) was as follows: S^2−^ > CO_3_^2−^ > Cl^−^ > HS^−^ > HCO_3_^−^.(4)With the addition of CO_3_^2−^ and S^2−^, local corrosion of the TC4 alloy in an NaCl solution containing saturated CO_2_ increased, especially the synergistic effect between Cl^−^, CO_3_^2−^, and/or S^2−^, which made the corrosion electrochemical parameters of TC4 alloy change by two orders of magnitude.

## Figures and Tables

**Figure 1 materials-16-03129-f001:**
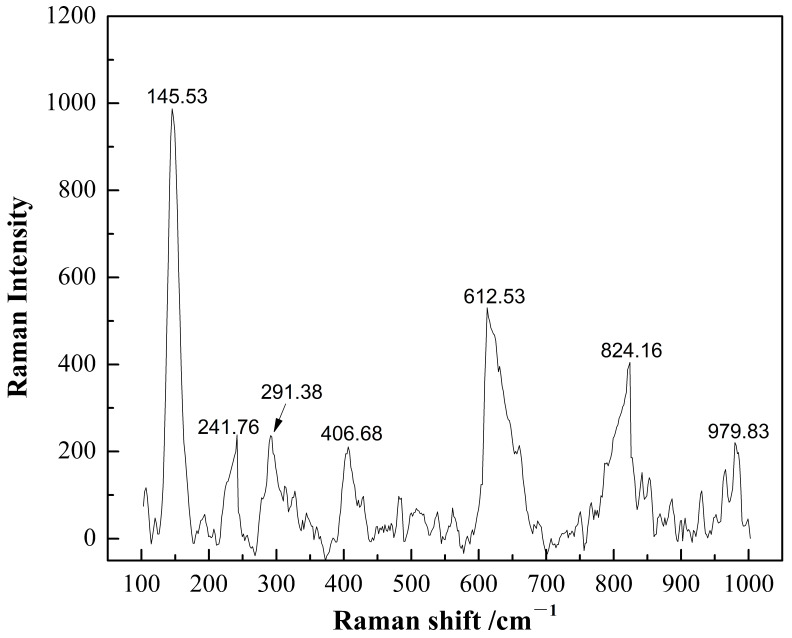
Raman spectra of TiO_2_ passivation film on TC4 titanium alloy surface.

**Figure 2 materials-16-03129-f002:**
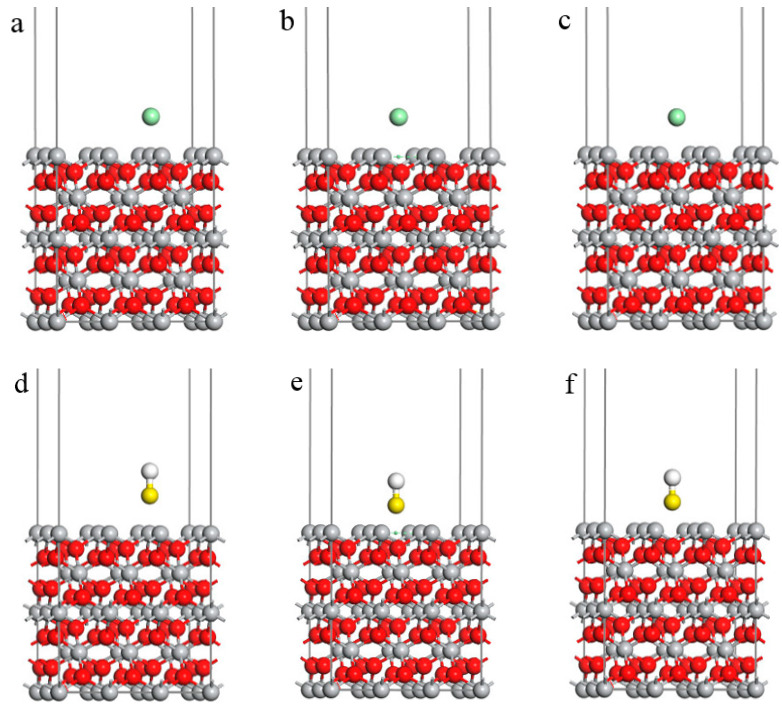
Interface model of TiO_2_100) including corrosive ions. (**a**) TiO_2_(100)-Cl^−^(top); (**b**) TiO_2_(100)-Cl^−^(bridge); (**c**) TiO_2_(100)-Cl^−^(hole); (**d**) TiO_2_(100)-HS^−^(top); (**e**) TiO_2_(100)-HS^−^(bridge); (**f**) TiO_2_(100)-HS^−^(hole); (**g**) TiO_2_(100)-S^2−^(top); (**h**) TiO_2_(100)-S^2−^(bridge); (**i**) TiO_2_(100)-S^2−^(hole); (**j**) TiO_2_(100)-HCO_3_^−^(top); (**k**) TiO_2_(100)-HCO_3_^−^(bridge); (**l**) TiO_2_(100)-HCO_3_^−^(hole); (**m**) TiO_2_(100)-CO_3_^2−^(top); (**n**) TiO_2_(100)-CO_3_^2−^(bridge); (**o**) TiO_2_(100)-CO_3_^2−^(hole).

**Figure 3 materials-16-03129-f003:**
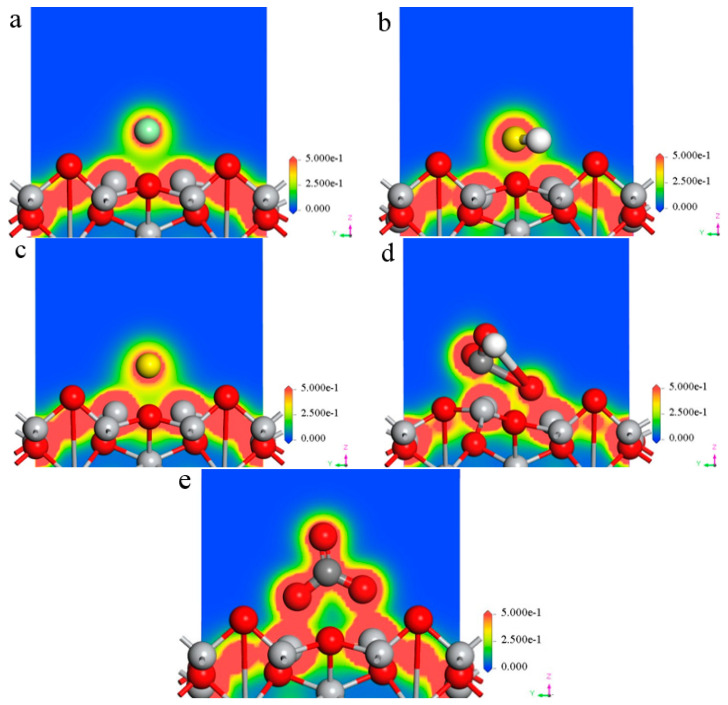
Charge density distribution of five ions on the TiO_2_(100) surface. (**a**) Cl^−^; (**b**) HS^−^; (**c**) S^2−^; (**d**) HCO_3_^−^; (**e**) CO_3_^2−^.

**Figure 4 materials-16-03129-f004:**
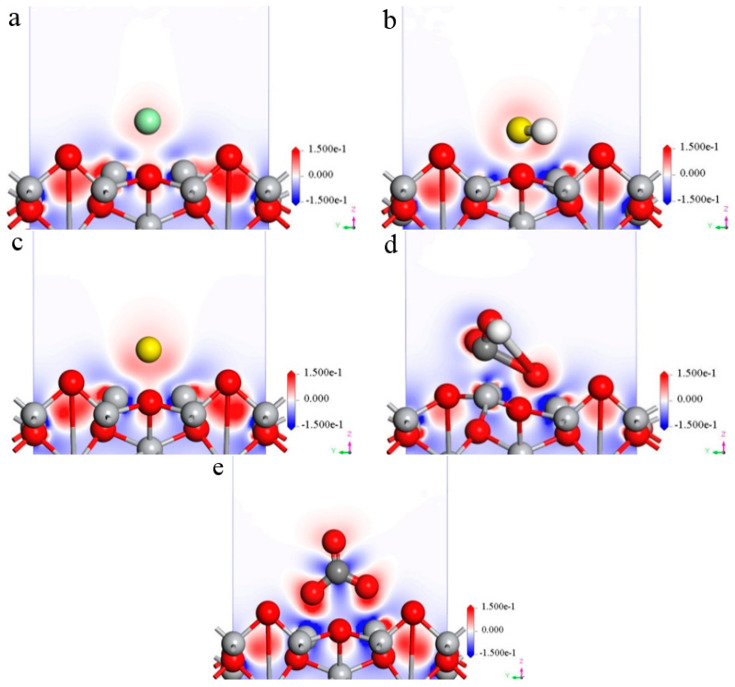
Differential charge density distribution of five ions on the TiO_2_(100) surface at the bridge site. (**a**) Cl^−^; (**b**) HS^−^; (**c**) S^2−^; (**d**) HCO_3_^−^; (**e**) CO_3_^2−^.

**Figure 5 materials-16-03129-f005:**
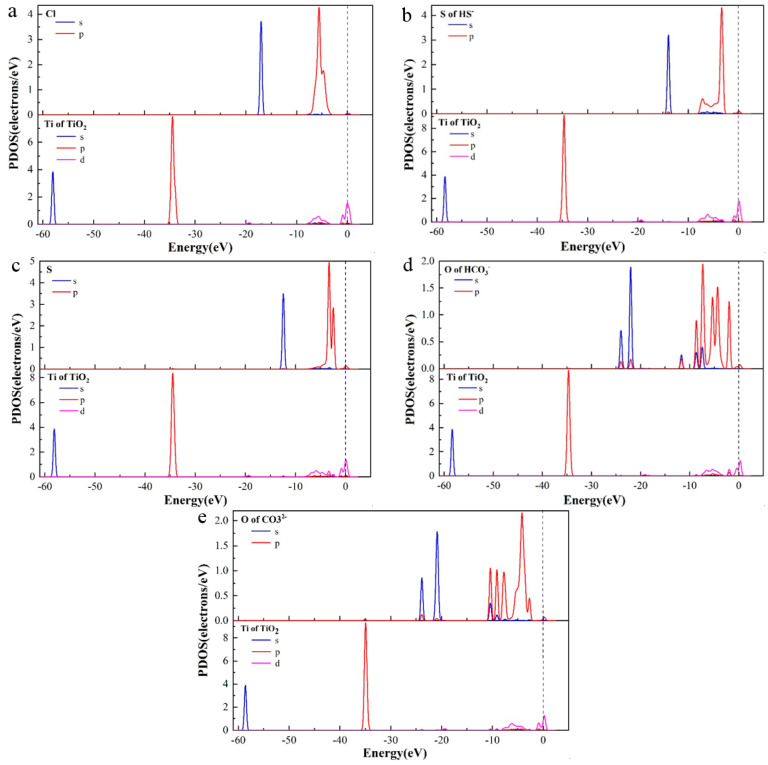
PDOS curves of (**a**) Cl^−^, (**b**) HS^−^, (**c**) S^2−^, (**d**) HCO_3_^−^, and (**e**) CO_3_^2−^ adsorbing on the TiO_2_(100) surface at the bridge site.

**Figure 6 materials-16-03129-f006:**
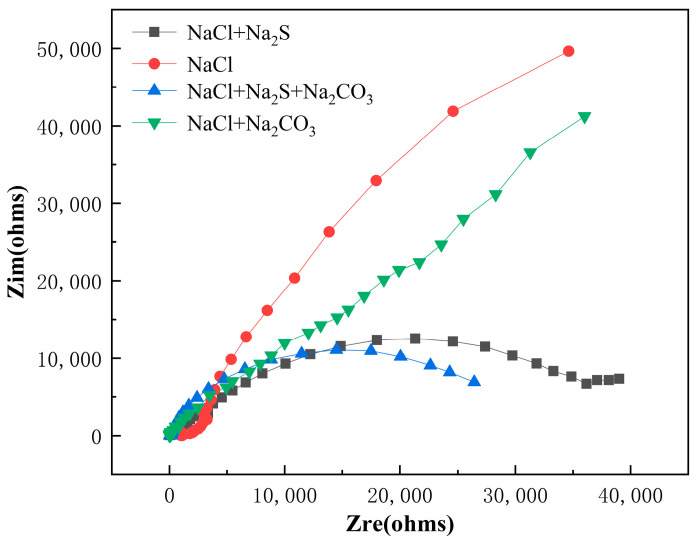
Alternating-current impedance diagram of TC4 titanium alloy under different corrosion environments.

**Figure 7 materials-16-03129-f007:**
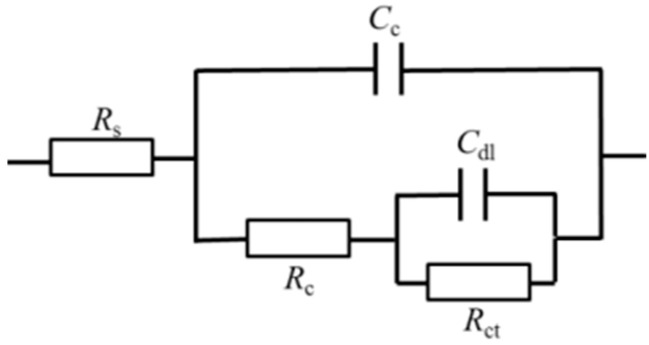
Equivalent circuit diagram.

**Figure 8 materials-16-03129-f008:**
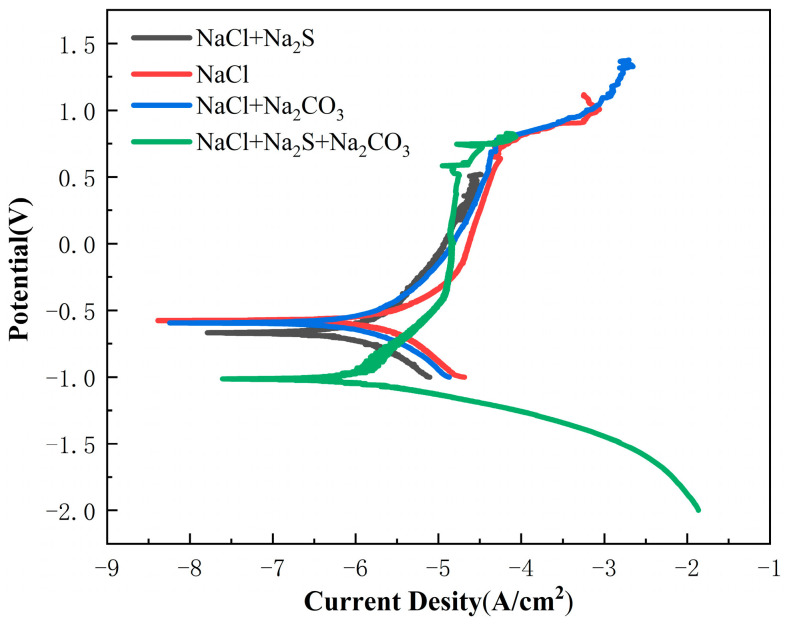
Polarization curves of TC4 titanium alloy under four corrosion environments.

**Table 1 materials-16-03129-t001:** Raman spectra characteristic frequency shift position of three crystalline TiO_2_.

Crystal Structure of TiO_2_	Raman Frequency Shift (cm^−1^)
Brookite	127, 150, 193, 212, 247, 286, 318, 366, 412, 462, 502, 544, 582, 645
Anatase	143, 196, 326, 395, 512, 645
Rutile	143, 244, 440, 610, 825

**Table 2 materials-16-03129-t002:** Final energy of five corrosive ions on the TiO_2_(100) surface at different sites.

Models	Final Energy/eV
TiO_2_(top)-Cl^−^	−69,578.2607299930
TiO_2_(bridge)-Cl^−^	−69,578.6903762455
TiO_2_(hole)-Cl^−^	−69,578.6028239061
TiO_2_(top)-HS^−^	−69,462.8589004437
TiO_2_(bridge)-HS^−^	−69,463.9047289728
TiO_2_(hole)-HS^−^	−69,463.8965710229
TiO_2_(top)-S^2−^	−69,443.5255285524
TiO_2_(bridge)-S^2−^	−69,445.7491433063
TiO_2_(hole)-S^2−^	−69,445.3972256848
TiO_2_(top)-HCO_3_^−^	−70,647.8573602164
TiO_2_(bridge)-HCO_3_^−^	−70,651.6819378591
TiO_2_(hole)-HCO_3_^−^	−70,649.4503215808
TiO_2_(top)-CO_3_^2−^	−70,633.5143850651
TiO_2_(bridge)-CO_3_^2−^	−70,633.5232150766
TiO_2_(hole)-CO_3_^2−^	−70,633.5217354935

**Table 3 materials-16-03129-t003:** Charge numbers of Cl, O, and S atoms in corrosive ions.

Atom	Cl^−^(Cl)	HCO_3_^−^(O)	CO_3_^2−^(O)	HS^−^(S)	S^2−^(S)
Charge/e	−0.15	−0.12	−0.22	−0.14	−0.34

**Table 4 materials-16-03129-t004:** The final energy of five ions, TiO_2_, and TiO_2_(100).

Model	Final Energy/eV
Cl^−^	−411.7437852366
HCO_3_^−^	−1483.4813928615
CO_3_^2−^	−1467.9443448936
HS^−^	−296.3156703881
S^2−^	−280.4987150773
TiO_2_	−4961.8922752454
TiO_2_(100)	−69171.5619281723

**Table 5 materials-16-03129-t005:** Interface binding energies of five ions on TiO_2_(100) surface at bridge sites.

Model	Interface Binding Energy/eV
TiO_2_(bridge)-Cl^−^	4.6153371666
TiO_2_(bridge)-HS^−^	3.9728695881
TiO_2_(bridge)-S^2−^	6.3114999473
TiO_2_(bridge)-HCO_3_^−^	3.3613831813
TiO_2_(bridge)-CO_3_^2−^	5.9830579936

**Table 6 materials-16-03129-t006:** Alternating-current fitting results of TC4 titanium alloy in various corrosion environments.

Corrosion Environments	*R*_s_ */*Ω·cm^2^	*C*_c_ × 10^−7^ /F·cm^−2^	*R*_c_ */*Ω·cm^2^	*C*_dl_ × 10^−6^ /F·cm^−2^	*R*_ct_ × 10^4^ */*Ω·cm^2^
NaCl	68.12 ± 0.72	3.617 ± 0.035	1565 ± 33	4.925 ± 0.062	3.135 ± 0.039
NaCl + Na_2_CO_3_	54.38 ± 0.55	3.826 ± 0.029	1324 ± 15	5.236 ± 0.051	2.754 ± 0.058
NaCl + Na_2_S	33.05 ± 0.61	4.182 ± 0.042	1069 ± 24	56.480 ± 0.038	2.376 ± 0.066
NaCl + Na_2_S + Na_2_CO_3_	9.923 ± 0.58	8.793 ± 0.053	948 ± 16	56.641 ± 0.042	0.633 ± 0.045

**Table 7 materials-16-03129-t007:** Electrochemical parameters of TC4 titanium alloy in various corrosion environments.

Corrosion Media	*E*_corr_ /mV	*b*_c_ /mV	*b*_a_ /mV	*i*_corr_ × 10^−4^ /mA·cm^−2^
NaCl	−578 ± 3.56	35.704 ± 0.086	29.744 ± 0.066	1.689 ± 0.034
NaCl + Na_2_CO_3_	−595 ± 2.35	33.951 ± 0.092	41.011 ± 0.085	9.399 ± 0.042
NaCl + Na_2_S	−667 ± 4.62	27.320 ± 0.075	27.218 ± 0.073	19.173 ± 0.071
NaCl + Na_2_S + Na_2_CO_3_	−1014 ± 4.85	22.659 ± 0.088	32.778 ± 0.069	252.519 ± 0.099

**Table 8 materials-16-03129-t008:** Electrochemical parameters of titanium alloy in 35% NaCl + 0.4% Na_2_S [25].

Temperature/°C	Materials	*R*_s_/Ω·cm^2^	*R*_ct_/Ω·cm^2^	*E*_corr/_V	*i*_corr_× 10^−4^ /mA·cm^−2^
25	TC4	62.49	947,740	−0.596	1.751
	TC4ELI	9.15	290,050	−0.553	2.468
80	TC4	15.52	62,373	−0.546	5.651
	TC4ELI	6.15	93,617	−0.402	4.105

## Data Availability

The data presented in this study are available on request from the corresponding author.

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
