# Peer review of "Adsorption Characteristics between Ti Atoms of TiO2(100) and Corrosive Species of CO2-H2S-Cl System in Oil and Gas Fields"

_materials, 2023, doi:10.3390/ma16083129_

Round 1

Reviewer 1 Report

I cannot recommend this paper for publication in its present form. Here are my remarks:

1- In Introduction, the first two paragraphs are the same. One of them should be removed.

2- Improvement in English is required. There are trivial grammatical errors in the text. (E.g., lines 94-95, "were studies" should be corrected as "were studied". Line 197, "show" should be corrected as "shows", etc.) Please check all the text carefully.

3- Line 197,  "PDOS" abbreviation should be introduced in parentheses immediately after the first use in the text.

4- The authors should be consistent on the use of terms related to corrosion. Do not use the terms of "erosive", "eroded, and "eroding" thoroughout the text!

5- Is the k-value sufficient for (2x3x1) surface supercell, because TC4 is an alloy? Also, I wonder if  the magnetic moment of two plates were constrained. Moreover, I think that 2-plate slab is too thin to yield converged adsorption energies. Authors therefore need to show that their
final energies (Tables 2 & 5) are converged with respect to k-value and thickness of the 2-layer slab.

6- Is there a specific reason to express all reported energy values (Table 2, 4 & 5) with 8 and 10 significant figures after the decimal point? 

7- In section 2.2.2, the authors should state the number of eletrochemical tests performed. Also, the standard deviation and scattering bars should be reported in Tables 6 & 7.

Author Response

Dear Ms. Irina Mariana Sandulache and reviewer:

Thank you for your letter and the reviewer’s comments on our manuscript entitled" Adsorption characteristics between Ti atoms of TiO2(100) and corrosive species of CO2-H2S-Cl- system in oil and gas fields". Those comments are very helpful for revising and improving our paper, as well as the important guiding significance to other research. We have studied the comments carefully and made corrections which we hope meet with approval. The main corrections are in the manuscript, and the responds to the reviewer’s comments are as follows (the replies are highlighted in red).

Replies to the reviewer’s comments:

1- In Introduction, the first two paragraphs are the same. One of them should be removed.

Response: We are very sorry for this wrong version. According to your advices, one of them has been removed. Please see Page 2.

2- Improvement in English is required. There are trivial grammatical errors in the text. (E.g., lines 94-95, "were studies" should be corrected as "were studied". Line 197, "show" should be corrected as "shows", etc.) Please check all the text carefully.

Response: According to your advices, the corresponding contents have been revised and were highlighted in red, and all the text have been checked carefully too.

3- Line 197, "PDOS" abbreviation should be introduced in parentheses immediately after the first use in the text.

Response: PDOS (Projected or Partial Density of States) further qualifies these results by resolving the contributions according to the angular momentum of the states. It is often useful to know whether the main peaks in the DOS are of s, p, or d character. PDOS calculations are based on Mulliken population analysis, which allows the contribution from each energy band to a given atomic orbital to be calculated. Now it has been added in the revised manuscript. Please see 1st paragraph from bottom in Page 7.

4- The authors should be consistent on the use of terms related to corrosion. Do not use the terms of "erosive", "eroded, and "eroding" thorough out the text!

Response: According to your advices, the use of terms related to corrosion thorough out the text has been revised.

5- Is the k-value sufficient for (2×3×1) surface supercell, because TC4 is an alloy? Also, I wonder if the magnetic moment of two plates were constrained. Moreover, I think that 2-plate slab is too thin to yield converged adsorption energies. Authors therefore need to show that their final energies (Tables 2 & 5) are converged with respect to k-value and thickness of the 2-layer slab.

Response: 2*3*1 is already a supercell structure.

The functionals of the exchange correlation GGA with exchange correlation of revised the Pardew–Burke–Ernzerhof (RPBE) for geometry relation were used. A 2 × 2 × 1 supercell was modeled with 48 atoms to perform the calculations. Furthermore, 3 × 3 × 1 k-points were used for the geometry optimization.

[1] Akbar Hussain, Abdur Rauf, Ejaz Ahmed, Muhammad Saleem Khan, Shabeer Ahmad Mian, Joonkyung Jang. Modulating Optoelectronic and Elastic Properties of Anatase TiO2 for Photoelectrochemical Water Splitting. Molecules, 2023, 28(7): 3252.

Different AFM spin configurations have been modeled within the unit cell (1 × 1 × 1) and for supercells extended out-of-plane (1 × 1 × 2) and in-plane (2 × 2 × 1).

[2] Quanzheng Tao, Joseph Halim, Justinas Palisaitis, Adam Carlsson, Martin Dahlqvist, Ulf Wiedwald, Michael Farle, Per O. Å. Persson, Johanna Rosen. Synthesis, Characterization, and Modeling of a Chemically Ordered Quaternary Boride, Mo4MnSiB2. Cryst. Growth Des. 2023, doi.org/10.1021/acs.cgd.2c01416.

Anatase TiO2 belongs to the tetragonal crystal system. The space group is I41/amd, and its vector lattice contains eight O atoms and four Ti atoms. Further, considering computing power and actual situation, a supercell model comprising four (2×3×1) original cells.

[3] Yonghong Gu, Congzhong Cai, Qing Feng, Yanhua Li. Spectrum redshift effect of anatase TiO2 codoped with nitrogen and first transition elements. Chinese Optics Letters, 2014: 12(9): 091602.

6- Is there a specific reason to express all reported energy values (Table 2, 4 & 5) with 8 and 10 significant figures after the decimal point?

Response: All data automatically were exported by the system. Now all reported energy values (Table 2, 4 & 5) were kept with 10 significant figures after the decimal point. Please see Tables 2, 4 & 5.

7- In section 2.2.2, the authors should state the number of eletrochemical tests performed. Also, the standard deviation and scattering bars should be reported in Tables 6 & 7.

Response: The number of eletrochemical tests performed were 3 times, as shown in section 2.2.2, and the standard deviations were reported in Tables 6 & 7。

Once again, thank you very much for your constructive comments and suggestions which would help us both in English and in depth to improve the quality of the paper.

Kind regards,

Shidong Zhu

zhusdxt@126.com

Reviewer 2 Report

Title: Adsorption characteristics between Ti atoms of TiO2(100) and corrosive species of CO2-H2S-Cl- system in oil and gas fields
Ref. No.: materials-2248124

Reviewer Comments:

This study tried to study Adsorption characteristics between Ti atoms of TiO2(100) and corrosive species of CO2-H2S-Cl- a system in oil and gas fields. This topic is quite interesting and in line with the research objectives of the Materials Journal.  So, I think it will be helpful to this journal. However, this paper has some critical problems and needs to be corrected to accept.

1.    The novelty and the main issue that want to be solved in this work must be addressed in the abstract and introduction. What is the main problem issue with the current electrode? The optimum CV values should be stated.

2.    English language should be carefully checked.

3.    Introduction section must be written in a more quality way, i.e. more up-to-date references addressed. The research gap should be delivered more clearly with the directed necessity for the conducted research work. For example,

a.    What is the problem with conventional titanium?

b.    Described why choose this material?

c.     The problem statement of this study is not so clear.

d.    The previous study should be discussed to highlight the significance of this study.

4.    The parameter of the study should be stated in the methodology. The condition of the instrument of characterization and parameter test should be stated.

5.    The discussion in section 3.1, the role of all materials used should be explained. How do the different sizes of TC4 significantly to results?

6.    Any mechanism explanation for material the role in this application? How this material propose is useful to improve it? 

7.    The comparison performance of electrodes for this study must do with previous studies and add one of the tables.

8.    The conclusion part should be revised to align the experimental and simulation part.

Author Response

Dear Ms. Irina Mariana Sandulache and reviewer:

Thank you for your letter and the reviewer’s comments on our manuscript entitled" Adsorption characteristics between Ti atoms of TiO2(100) and corrosive species of CO2-H2S-Cl- system in oil and gas fields". Those comments are very helpful for revising and improving our paper, as well as the important guiding significance to other research. We have studied the comments carefully and made corrections which we hope meet with approval. The main corrections are in the manuscript and the responds to the reviewer’s comments are as follows (the replies are highlighted in red).

Replies to the reviewer’s comments:

  1. The novelty and the main issue that want to be solved in this work must be addressed in the abstract and introduction. What is the main problem issue with the current electrode? The optimum CV values should be stated.

Response: According to your advices, the Abstract section has been rewritten.

  1. English language should be carefully checked.

Response: According to your advices, English language has been carefully checked. Please see the corresponding contents highlighted in red in the revised manuscript.

  1. Introduction section must be written in a more quality way, i.e. more up-to-date references addressed. The research gap should be delivered more clearly with the directed necessity for the conducted research work. For example,
  2. What is the problem with conventional titanium?
  3. Described why choose this material?
  4. The problem statement of this study is not so clear.
  5. The previous study should be discussed to highlight the significance of this study.

Response: According to your advices, the Introduction section has been rewritten.

  1. The parameter of the study should be stated in the methodology. The condition of the instrument of characterization and parameter test should be stated.

Response: The parameters (Temperature, solution, equipment etc) of the study have be stated in 2.2.2. Experimental methods and equipment.

  1. The discussion in section 3.1, the role of all materials used should be explained. How do the different sizes of TC4 significantly to results?

Response: The aim is to simulate the species in CO2-H2S-Cl- environment, which has been added in 3.1.1 Section in Page 5.

CO2+H2O→H2CO3

H2CO3→H++HCO3-

HCO3-→H++CO32-

H2S→H++HS-

HS-→H++S2-

  1. Any mechanism explanation for material the role in this application? How this material propose is useful to improve it?

Response: The presence of S2- in NaCl solution, S2- will react with Ti4+ and product TiS2, indicating that TiO2 film is destroyed, so the corrosion resistance of TC4 alloy to corrosive species in CO2-H2S-Cl- environment decrease.

Therefore, spraying a film on the surface of the titanium alloy or adding the corresponding corrosion inhibitor to decrease the biding energy between S from corrosive species and Ti in TiO2 will enhance the corrosion resistance of TC4 alloy.

  1. The comparison performance of electrodes for this study must do with previous studies and add one of the tables.

Response: According to your advices, Table 8 was added as follows. Compared with the 3.5% NaCl +0.4% Na2S at the same temperature 80 ℃。

Alternating-current fitting results of TC4 titanium alloy in various corrosion environments.

Corrosion environments

Rs

/Ω·cm2

Cc×10-7

/F·cm-2

Rc

/Ω·cm2

Cdl×10-6

/F·cm-2

Rct×104

/Ω·cm2

NaCl

68.12±0.72

3.617±0.035

1565±33

4.925±0.062

3.135±0.039

NaCl+Na2CO3

54.38±0.55

3.826±0.029

1324±15

5.236±0.051

2.754±0.058

NaCl+Na2S

33.05±0.61

4.182±0.042

1069±24

56.480±0.038

2.376±0.066

NaCl+Na2S+Na2CO3

9.923±0.58

8.793±0.053

948±16

56.641±0.042

0.633±0.045

Electrochemical parameters of TC4 titanium alloy in various corrosion environments.

Corrosion media

Ecorr

/mV

bc

/mV

ba

/mV

icorr×10-4

/mA·cm-2

NaCl

-578±3.56

35.704±0.086

29.744±0.066

1.689±0.034

NaCl+Na2CO3

-595±2.35

33.951±0.092

41.011±0.085

9.399±0.042

NaCl+Na2S

-667±4.62

27.320±0.075

27.218±0.073

19.173±0.071

NaCl+Na2S+Na2CO3

-1014±4.85

22.659±0.088

32.778±0.069

252.519±0.099

Table 8. Electrochemical parameters of titanium ally in 35% NaCl + 0.4% Na2S [25]

Temperature /oC

Materials

Rs/Ω·cm2

Rct /Ω·cm2

Ecorr /V

icorr /A·cm-2

25

TC4

62.49

947740

-0.596

1.751e-7

TC4ELI

9.15

290050

-0.553

2.4e-7

80

TC4

15.52

62373

-0.546

5.651e-7

TC4ELI

6.15

93617

-0.402

2.4e-7

[25] Chen, C.; Shuang, Y.H.; Chen, J.X.; Zhang, Y.Z; Liu B.S.; Zhang, S.H. Research on electrochemical characteristics and material selection of corrosion resistant titanium alloy in petroleum field. Light Metals (in Chinese), 2023,(1): 54-59.

  1. The conclusion part should be revised to align the experimental and simulation part.

Response: According to your advices, the Conclusions section has been revised. Please see them in Page 11.

Once again, thank you very much for your constructive comments and suggestions which would help us both in English and in depth to improve the quality of the paper.

Kind regards,

Shidong Zhu

zhusdxt@126.com

Round 2

Reviewer 1 Report

The manuscript has been revised according to my previous comments, and thus, it can now be accepted for publication.